# Eigenvalue Decay Implies Polynomial-Time Learnability for Neural Networks

**Surbhi Goel** [*]
Department of Computer Science
University of Texas at Austin
surbhi@cs.utexas.edu

**Adam Klivans** [†]
Department of Computer Science
University of Texas at Austin
klivans@cs.utexas.edu

## Abstract

We consider the problem of learning function classes computed by neural networks with various activations (e.g. ReLU or Sigmoid), a task believed to be computationally intractable in the worst-case. A major open problem is to understand the minimal assumptions under which these classes admit provably efficient algorithms. In this work we show that a natural distributional assumption corresponding to *eigenvalue decay* of the Gram matrix yields polynomial-time algorithms in the non-realizable setting for expressive classes of networks (e.g. feed-forward networks of ReLUs). We make no assumptions on the structure of the network or the labels. Given sufficiently-strong eigenvalue decay, we obtain *fully*-polynomial time algorithms in *all* the relevant parameters with respect to square-loss. This is the first purely distributional assumption that leads to polynomial-time algorithms for networks of ReLUs. Further, unlike prior distributional assumptions (e.g., the marginal distribution is Gaussian), eigenvalue decay has been observed in practice on common data sets.

## 1 Introduction

Understanding the computational complexity of learning neural networks from random examples is a fundamental problem in machine learning. Several researchers have proved results showing computational *hardness* for the worst-case complexity of learning various networks– that is, when no assumptions are made on the underlying distribution or the structure of the network [10, 16, 21, 26, 43]. As such, it seems necessary to take some assumptions in order to develop efficient algorithms for learning deep networks (the most expressive class of networks known to be learnable in polynomial-time without any assumptions is a sum of one hidden layer of sigmoids [16]). A major open question is to understand what are the "correct" or minimal assumptions to take in order to guarantee efficient learnability[3]. An oft-taken assumption is that the marginal distribution is equal to some smooth distribution such as a multivariate Gaussian. Even under such a distributional assumption, however, there is evidence that fully polynomial-time algorithms are still hard to obtain for simple classes of networks [19, 36]. As such, several authors have made further assumptions on the underlying structure of the model (and/or work in the noiseless or *realizable* setting).

In fact, in an interesting recent work, Shamir [34] has given evidence that both distributional assumptions and assumptions on the network structure are necessary for efficient learnability using gradient-based methods. Our main result is that under *only* an assumption on the marginal distribution, namely eigenvalue decay of the Gram matrix, there exist efficient algorithms for learning broad

---

[*]Work supported by a Microsoft Data Science Initiative Award.

[†]Part of this work was done while visiting the Simons Institute for Theoretical Computer Science.

[3]For example, a very recent paper of Song, Vempala, Xie, and Williams [36] asks "What form would such an explanation take, in the face of existing complexity-theoretic lower bounds?"

classes of neural networks even in the non-realizable (agnostic) setting with respect to square loss. Furthermore, eigenvalue decay has been observed often in real-world data sets, unlike distributional assumptions that take the marginal to be unimodal or Gaussian. As one would expect, stronger assumptions on the eigenvalue decay result in polynomial learnability for broader classes of networks, but even mild eigenvalue decay will result in savings in runtime and sample complexity.

The relationship between our assumption on eigenvalue decay and prior assumptions on the marginal distribution being Gaussian is similar in spirit to the dichotomy between the complexity of certain algorithmic problems on power-law graphs versus Erdős-Rényi graphs. Several important graph problems such as clique-finding become much easier when the underlying model is a random graph with appropriate power-law decay (as opposed to assuming the graph is generated from the classical $G(n, p)$ model) [6, 22]. In this work we prove that neural network learning problems become tractable when the underlying distribution induces an empirical gram matrix with sufficiently strong eigenvalue-decay.

**Our Contributions.** Our main result is quite general and holds for any function class that can be suitably embedded in an RKHS (Reproducing Kernel Hilbert Space) with corresponding kernel function $k$ (we refer readers unfamiliar with kernel methods to [30]). Given $m$ draws from a distribution $(\mathbf{x}_1, \ldots, \mathbf{x}_m)$ and kernel $k$, recall that the *Gram matrix $K$* is an $m \times m$ matrix where the $i, j$ entry equals $k(\mathbf{x}_i, \mathbf{x}_j)$. For ease of presentation, we begin with an informal statement of our main result that highlights the relationship between the eigenvalue decay assumption and the run-time and sample complexity of our final algorithm.

**Theorem 1** (Informal). *Fix function class $\mathcal{C}$ and kernel function $k$. Assume $\mathcal{C}$ is approximated in the corresponding RKHS with norm bound $B$. After drawing $m$ samples, let $K/m$ be the (normalized) $m \times m$ Gram matrix with eigenvalues $\{\lambda_1, \ldots, \lambda_m\}$. For error parameter $\epsilon > 0$,*

*1. If, for sufficiently large $i$, $\lambda_i \approx O(i^{-p})$, then $\mathcal{C}$ is efficiently learnable with $m = \tilde{O}(B^{1/p}/\epsilon^{2+3/p})$.*

*2. If, for sufficiently large $i$, $\lambda_i \approx O(e^{-i})$, then $\mathcal{C}$ is efficiently learnable with $m = \tilde{O}(\log B/\epsilon^2)$.*

We allow a failure probability for the event that the eigenvalues do not decay. In all prior work, the sample complexity $m$ depends linearly on $B$, and for many interesting concept classes (such as ReLUs), $B$ is exponential in one or more relevant parameters. Given Theorem 1, we can use known structural results for embedding neural networks into an RKHS to estimate $B$ and take a corresponding eigenvalue decay assumption to obtain polynomial-time learnability. Applying bounds recently obtained by Goel et al. [16] we have

**Corollary 2.** *Let $\mathcal{C}$ be the class of all fully-connected networks of ReLUs with one-hidden layer of $\ell$ hidden ReLU activations feeding into a single ReLU output activation (i.e., two hidden layers or depth-3). Then, assuming eigenvalue decay of $O(i^{-\ell/\epsilon})$, $\mathcal{C}$ is learnable in polynomial time with respect to square loss on $\mathbb{S}^{n-1}$. If ReLU is replaced with sigmoid, then we require eigenvalue decay $O(i^{-\sqrt{\ell}\log(\sqrt{\ell}/\epsilon)})$.*

For higher depth networks, bounds on the required eigenvalue decay can be derived from structural results in [16]. Without taking an assumption, the fastest known algorithms for learning the above networks run in time exponential in the number of hidden units and accuracy parameter (but polynomial in the dimension) [16].

Our proof develops a novel approach for bounding the generalization error of kernel methods, namely we develop *compression schemes* tailor-made for classifiers induced by kernel-based regression, as opposed to current Rademacher-complexity based approaches. Roughly, a compression scheme is a mapping from a training set $S$ to a small subsample $S'$ and *side-information $\mathcal{I}$*. Given this compressed version of $S$, the decompression algorithm should be able to generate a classifier $h$. In recent work, David, Moran and Yehudayoff [13] have observed that if the size of the compression is much less than $m$ (the number of samples), then the empirical error of $h$ on $S$ is close to its true error with high probability.

At the core of our compression scheme is a method for giving small description length (i.e., $o(m)$ bit complexity), approximate solutions to instances of kernel ridge regression. Even though we assume $K$ has decaying eigenvalues, $K$ is neither sparse nor low-rank, and even a single column or row of $K$ has bit complexity at least $m$, since $K$ is an $m \times m$ matrix! Nevertheless, we can prove that recent tools from Nyström sampling [28] imply a type of sparsification for solutions

of certain regression problems involving $K$. Additionally, using preconditioning, we can bound the bit complexity of these solutions and obtain the desired compression scheme. At each stage we must ensure that our compressed solutions do not lose too much accuracy, and this involves carefully analyzing various matrix approximations. Our methods are the first compression-based generalization bounds for kernelized regression.

**Related Work.** Kernel methods [30] such as SVM, kernel ridge regression and kernel PCA have been extensively studied due to their excellent performance and strong theoretical properties. For large data sets, however, many kernel methods become computationally expensive. The literature on approximating the Gram matrix with the overarching goal of reducing the time and space complexity of kernel methods is now vast. Various techniques such as random sampling [39], subspace embedding [2], and matrix factorization [15] have been used to find a low-rank approximation that is efficient to compute and gives small approximation error. The most relevant set of tools for our paper is Nyström sampling [39, 14], which constructs an approximation of $K$ using a subset of the columns indicated by a selection matrix $S$ to generate a positive semi-definite approximation. Recent work on leverage scores have been used to improve the guarantees of Nyström sampling in order to obtain linear time algorithms for generating these approximations [28].

The novelty of our approach is to use Nyström sampling in conjunction with compression schemes to give a new method for giving provable *generalization error* bounds for kernel methods. Compression schemes have typically been studied in the context of classification problems in PAC learning and for combinatorial problems related to VC dimension [23, 24]. Only recently some authors considered compression schemes in a general, real-valued learning scenario [13]. Cotter, Shalev-Shwartz, and Srebro have studied compression for classification using SVMs to prove that for general distributions, compressing classifiers with low generalization error is not possible [9].

The general phenomenon of eigenvalue decay of the Gram matrix has been studied from both a theoretical and applied perspective. Some empirical studies of eigenvalue decay and related discussion can be found in [27, 35, 38]. There has also been prior work relating eigenvalue decay to generalization error in the context of SVMs or Kernel PCA (e.g., [29, 35]). Closely related notions to eigenvalue decay are that of *local Rademacher complexity* due to Bartlett, Bousquet, and Mendelson [4] (see also [5]) and that of *effective dimensionality* due to Zhang [42].

The above works of Bartlett et al. and Zhang give improved generalization bounds via data-dependent estimates of eigenvalue decay of the kernel. At a high level, the goal of these works is to work under an assumption on the effective dimension and improve Rademacher-based generalization error bounds from $1/\sqrt{m}$ to $1/m$ ($m$ is the number of samples) for functions embedded in a RKHS of unit norm. These works do not address the main obstacle of this paper, however, namely overcoming the complexity of the norm of the approximating RKHS. Their techniques are mostly incomparable even though the intent of using effective dimension as a measure of complexity is the same.

Shamir has shown that for general linear prediction problems with respect to square-loss and norm bound $B$, a sample complexity of $\Omega(B)$ is required for gradient-based methods [33]. Our work shows that eigenvalue decay can dramatically reduce this dependence, even in the context of kernel regression where we want to run in time polynomial in $n$, the dimension, rather than the (much larger) dimension of the RKHS.

**Recent work on Learning Neural Networks.** Due in part to the recent exciting developments in deep learning, there have been several works giving provable results for learning neural networks with various activations (threshold, sigmoid, or ReLU). For the most part, these results take various assumptions on either 1) the distribution (e.g., Gaussian or Log-Concave) or 2) the structure of the network architecture (e.g. sparse, random, or non-overlapping weight vectors) or both and often have a bad dependence on one or more of the relevant parameters (dimension, number of hidden units, depth, or accuracy). Another way to restrict the problem is to work only in the noiseless/realizable setting. Works that fall into one or more of these categories include [20, 44, 40, 17, 31, 41, 11]. Kernel methods have been applied previously to learning neural networks [43, 26, 16, 12]. The current broadest class of networks known to be learnable in fully polynomial-time in all parameters with no assumptions is due to Goel et al. [16], who showed how to learn a sum of one hidden layer of sigmoids over the domain of $\mathbb{S}^{n-1}$, the unit sphere in $n$ dimensions. We are not aware of other prior

work that takes only a distributional assumption on the marginal and achieves fully polynomial-time algorithms for even simple networks (for example, one hidden layer of ReLUs).

Much work has also focused on the ability of gradient descent to succeed in parameter estimation for learning neural networks under various assumptions with an intense focus on the structure of local versus global minima [8, 18, 7, 37]. Here we are interested in the traditional task of learning in the non-realizable or agnostic setting and allow ourselves to output a hypothesis outside the function class (i.e., we allow improper learning). It is well known that for even simple neural networks, for example for learning a sigmoid with respect to square-loss, there may be many bad local minima [1]. Improper learning allows us to avoid these pitfalls.

## 2 Preliminaries

**Notation.** The input space is denoted by $\mathcal{X}$ and the output space is denoted by $\mathcal{Y}$. Vectors are represented with boldface letters such as $\mathbf{x}$. We denote a kernel function by $k_\psi(x, x') = \langle \psi(x), \psi(x') \rangle$ where $\psi$ is the associated feature map and for the kernel and $\mathcal{K}_\psi$ is the corresponding reproducing kernel Hilbert space (RKHS). For necessary background material on kernel methods we refer the reader to [30].

**Selection and Compression Schemes.** It is well known that in the context of PAC learning Boolean function classes, a suitable type of compression of the training data implies learnability [25]. Perhaps surprisingly, the details regarding the relationship between compression and ceratin other real-valued learning tasks have not been worked out until very recently. A convenient framework for us will be the notion of compression and selection schemes due to David et al. [13].

A selection scheme is a pair of maps $(\kappa, \rho)$ where $\kappa$ is the selection map and $\rho$ is the reconstruction map. $\kappa$ takes as input a sample $\mathcal{S} = ((\mathbf{x}_1, y_1), \ldots, (\mathbf{x}_m, y_m))$ and outputs a sub-sample $\mathcal{S}'$ and a finite binary string $b$ as side information. $\rho$ takes this input and outputs a hypothesis $h$. The *size* of the selection scheme is defined to be $k(m) = |\mathcal{S}'| + |b|$. We present a slightly modified version of the definition of an approximate compression scheme due to [13]:

**Definition 3** (($\epsilon, \delta$)-approximate agnostic compression scheme). *A selection scheme $(\kappa, \rho)$ is an ($\epsilon, \delta$)-approximate agnostic compression scheme for hypothesis class $\mathcal{H}$ and sample satisfying property $P$ if for all samples $\mathcal{S}$ that satisfy $P$ with probability $1 - \delta$, $f = \rho(\kappa(S))$ satisfies $\sum_{i=1}^m l(f(\boldsymbol{x}_i), y_i) \leq \min_{h \in \mathcal{H}} \left( \sum_{i=1}^m l(h(\boldsymbol{x}_i), y_i) \right) + \epsilon$.*

Compression has connections to learning in the general loss setting through the following theorem which shows that as long as $k(m)$ is small, the selection scheme generalizes.

**Theorem 4** (Theorem 30.2 [32], Theorem 3.2 [13]). *Let $(\kappa, \rho)$ be a selection scheme of size $k = k(m)$, and let $A_\mathcal{S} = \rho(\kappa(\mathcal{S}))$. Given $m$ i.i.d. samples drawn from any distribution $\mathcal{D}$ such that $k \leq m/2$, for constant bounded loss function $l : \mathcal{Y}' \times \mathcal{Y} \to \mathbb{R}^+$ with probability $1 - \delta$, we have*

$$\left| E_{(\boldsymbol{x}, y) \sim \mathcal{D}}[l(A_\mathcal{S}(x), y)] - \sum_{i=1}^m l(A_\mathcal{S}(\boldsymbol{x}_i), y_i) \right| \leq \sqrt{\epsilon \cdot \left( \frac{1}{m} \sum_{i=1}^m l(A_\mathcal{S}(\boldsymbol{x}_i), y_i) \right)} + \epsilon$$

*where $\epsilon = 50 \cdot \frac{k \log(m/k) + \log(1/\delta)}{m}$.*

## 3 Problem Overview

In this section we give a general outline for our main result. Let $\mathcal{S} = \{(\mathbf{x}_1, y_1), \ldots, (\mathbf{x}_m, y_m)\}$ be a training set of samples drawn i.i.d. from some arbitrary distribution $\mathcal{D}$ on $\mathcal{X} \times [0, 1]$ where $\mathcal{X} \subseteq \mathbb{R}^n$. Let us consider a concept class $\mathcal{C}$ such that for all $c \in \mathcal{C}$ and $\mathbf{x} \in \mathcal{X}$ we have $c(\mathbf{x}) \in [0, 1]$. We wish to learn the concept class $\mathcal{C}$ with respect to the square loss, that is, we wish to find $c \in \mathcal{C}$ that approximately minimizes $E_{(\mathbf{x}, y) \sim \mathcal{D}}[(c(\mathbf{x}) - y)^2]$. A common way of solving this is by solving the empirical minimization problem (ERM) given below and subsequently proving that it generalizes.

**Optimization Problem 1**

$$\underset{c \in \mathcal{C}}{\text{minimize}} \quad \frac{1}{m}\sum_{i=1}^{m}(c(\mathbf{x}_i) - y_i)^2$$

Unfortunately, it may not be possible to efficiently solve the ERM in polynomial-time due to issues such as non-convexity. A way of tackling this is to show that the concept class can be approximately minimized by another hypothesis class of linear functions in a high dimensional feature space (this in turn presents new obstacles for proving generalization-error bounds, which is the focus of this paper).

**Definition 5** ($\epsilon$-approximation)**.** *Let $\mathcal{C}_1$ and $\mathcal{C}_2$ be function classes mapping domain $\mathcal{X}$ to $\mathbb{R}$. $\mathcal{C}_1$ is $\epsilon$-approximated by $\mathcal{C}_2$ if for every $c \in \mathcal{C}_1$ there exists $c' \in \mathcal{C}_2$ such that for all $x \in \mathcal{X}$, $|c(x) - c'(x)| \leq \epsilon$.*

Suppose $\mathcal{C}$ can be $\epsilon$-approximated in the above sense by the hypothesis class $H_\psi = \{\mathbf{x} \to \langle \mathbf{v}, \psi(\mathbf{x}) \rangle | \mathbf{v} \in \mathcal{K}_\psi, \langle \mathbf{v}, \mathbf{v} \rangle \leq B\}$ for some $B$ and kernel function $k_\psi$. We further assume that the kernel is bounded, that is, $|k_\psi(\mathbf{x}, \mathbf{x'})| \leq M$ for some $M > 0$ for all $\mathbf{x}, \mathbf{x'} \in \mathcal{X}$. Thus, the problem relaxes to the following,

**Optimization Problem 2**

$$\underset{v \in \mathcal{K}_\psi}{\text{minimize}} \quad \frac{1}{m}\sum_{i=1}^{m}(\langle \mathbf{v}, \psi(\mathbf{x}_i) \rangle - y_i)^2 \qquad \text{subject to} \qquad \langle \mathbf{v}, \mathbf{v} \rangle \leq B$$

Using the Represener theorem, we have that the optimum solution for the above is of the form $\mathbf{v}^* = \sum_{i=1}^{m} \alpha_i \psi(\mathbf{x}_i)$ for some $\alpha \in \mathbb{R}^n$. Denoting the sample kernel matrix be $K$ such that $K_{i,j} = k_\psi(\mathbf{x}_i, \mathbf{x}_j)$, the above optimization problem is equivalent to the following optimization problem,

**Optimization Problem 3**

$$\underset{\alpha \in \mathbb{R}^m}{\text{minimize}} \quad \frac{1}{m}||K\alpha - Y||_2^2 \qquad \text{subject to} \qquad \alpha^T K \alpha \leq B$$

where $Y$ is the vector corresponding to all $y_i$ and $||Y||_\infty \leq 1$ since $\forall i \in [m], y_i \in [0,1]$. Let $\alpha_B$ be the optimal solution of the above problem. This is known to be efficiently solvable in $\text{poly}(m,n)$ time as long as the kernel function is efficiently computable.

Applying Rademacher complexity bounds to $\mathcal{H}_\psi$ yields generalization error bounds that decrease, roughly, on the order of $B/\sqrt{m}$ (c.f. Supplemental 1.1). If $B$ is exponential in $1/\epsilon$, the accuracy parameter, or in $n$, the dimension, as in the case of bounded depth networks of ReLUs, then this dependence leads to exponential sample complexity. As mentioned in Section 1, in the context of eigenvalue decay, various results [42, 4, 5] have been obtained to improve the dependence of $B/\sqrt{m}$ to $B/m$, but little is known about improving the dependence on $B$.

Our goal is to show that eigenvalue decay of the empirical Gram matrix does yield generalization bounds with better dependence on $B$. The key is to develop a novel compression scheme for kernelized ridge regression. We give a step-by-step analysis for how to generate an approximate, compressed version of the solution to Optimization Problem 3. Then, we will carefully analyze the bit complexity of our approximate solution and realize our compression scheme. Finally, we can put everything together and show how quantitative bounds on eigenvalue decay directly translate into compressions schemes with low generalization error.

## 4 Compressing the Kernel Solution

Through a sequence of steps, we will sparsify $\alpha$ to find a solution of much smaller bit complexity that is still an approximate solution (to within a small additive error). The quality and size of the approximation will depend on the eigenvalue decay.

**Lagrangian Relaxation.** We relax Optimization Problem 3 and consider the Lagrangian version of the problem to account for the norm bound constraint. This version is convenient for us, as it has a nice closed-form solution.

---

**Optimization Problem 4**

---

$$\underset{\alpha \in \mathbb{R}^m}{\text{minimize}} \qquad \frac{1}{m}||K\alpha - Y||_2^2 + \lambda \alpha^T K \alpha$$

---

We will later set $\lambda$ such that the error of considering this relaxation is small. It is easy to see that the optimal solution for the above lagrangian version is $\alpha = (K + \lambda m I)^{-1} Y$.

**Preconditioning.** To avoid extremely small or non-zero eigenvalues, we consider a perturbed version of $K$, $K_\gamma = K + \gamma m I$. This gives us that the eigenvalues of $K_\gamma$ are always greater than or equal to $\gamma m$. This property is useful for us in our later analysis. Henceforth, we consider the following optimization problem on the perturbed version of K:

---

**Optimization Problem 5**

---

$$\underset{\alpha \in \mathbb{R}^m}{\text{minimize}} \qquad \frac{1}{m}||K_\gamma \alpha - Y||_2^2 + \lambda \alpha^T K_\gamma \alpha$$

---

The optimal solution for perturbed version is $\alpha_\gamma = (K_\gamma + \lambda m I)^{-1} Y = (K + (\lambda + \gamma)m I)^{-1} Y$.

**Sparsifying the Solution via Nyström Sampling.** We will now use tools from Nyström Sampling to sparsify the solution obtained from Optimzation Problem 5. To do so, we first recall the definition of effective dimension or degrees of freedom for the kernel [42]:

**Definition 6** ($\eta$-effective dimension)**.** *For a positive semidefinite $m \times m$ matrix $K$ and parameter $\eta$, the $\eta$-effective dimension of $K$ is defined as $d_\eta(K) = tr(K(K + \eta m I)^{-1})$.*

Various kernel approximation results have relied on this quantity, and here we state a recent result due to [28] who gave the first application independent result that shows that there is an efficient way of computing a set of columns of $K$ such that $\bar{K}$, a matrix constructed from the columns is close in terms of 2-norm to the matrix $K$. More formally,

**Theorem 7** ([28])**.** *For kernel matrix $K$, there exists an algorithm that gives a set of $O\left(d_\eta(K) \log\left(d_\eta(K)/\delta\right)\right)$ columns, such that $\bar{K} = KS(S^T K S)^\dagger S^T K$ where $S$ is the matrix that selects the specific columns, satisfies with probability $1 - \delta$, $\bar{K} \preceq K \preceq \bar{K} + \eta m I$.*

It can be shown that $\bar{K}$ is positive semi-definite. Also, the above implies $||K - \bar{K}||_2 \leq \eta m$. We use the decay to approximate the Kernel matrix with a low-rank matrix constructed using the columns of $K$. Let $\bar{K}_\gamma$ be the matrix obtained by applying Theorem 7 to $K_\gamma$ for $\eta > 0$ and consider the following optimization problem,

---

**Optimization Problem 6**

---

$$\underset{\alpha \in \mathbb{R}^m}{\text{minimize}} \qquad \frac{1}{m}||\bar{K}_\gamma \alpha - Y||_2^2 + \lambda \alpha^T \bar{K}_\gamma \alpha$$

---

The optimal solution for the above is $\bar{\alpha}_\gamma = \left(\bar{K}_\gamma + \lambda m I\right)^{-1} Y$. Since $\bar{K}_\gamma = K_\gamma S(S^T K_\gamma S)^\dagger S^T K_\gamma$, solving for the above enables us to get a solution $\alpha^* = S(S^T K_\gamma S)^\dagger S^T K_\gamma \bar{\alpha}_\gamma$, which is a $k$-sparse vector for $k = O\left(d_\eta(K_\gamma) \log\left(d_\eta(K_\gamma)/\delta\right)\right)$.

**Bounding the Error of the Sparse Solution.** We bound the additional error incurred by our sparse hypothesis $\alpha^*$ compared to $\alpha_B$. To do so, we bound the error for each of the approximations: sparsification, preconditioning and lagrangian relaxation and then combine them to give the following theorem.

**Theorem 8** (Total Error)**.** *For $\lambda = \frac{\epsilon^2}{81B}$, $\eta \leq \frac{\epsilon^3}{729B}$ and $\gamma \leq \frac{\epsilon^3}{729B}$, we have $\frac{1}{m}||K_\gamma \alpha^* - Y||_2^2 \leq \frac{1}{m}||K\alpha_B - Y||_2^2 + \epsilon$.*

**Computing the Sparsity of the Solution.** To compute the sparsity of the solution, we need to bound $d_\eta(K_\beta)$. We consider the following different eigenvalue decays.

**Definition 9** (Eigenvalue Decay). *Let the real eigenvalues of a symmetric $m \times m$ matrix $A$ be denoted by $\lambda_1 \geq \cdots \geq \lambda_m$.*

*1. $A$ is said to have $(C, p)$-**polynomial eigenvalue decay** if for all $i \in \{1, \ldots, m\}$, $\lambda_i \leq Ci^{-p}$.*

*2. $A$ is said to have $C$-**exponential eigenvalue decay** if for all $i \in \{1, \ldots, m\}$, $\lambda_i \leq Ce^{-i}$.*

Note that in the above definitions $C$ and $p$ are not necessarily constants. We allow $C$ and $p$ to depend on other parameters (the choice of these parameters will be made explicit in subsequent theorem statements). We can now bound the effective dimension in terms of eigenvalue decay:

**Theorem 10** (Bounding effective dimension). *For $\gamma m \leq \eta$,*

*1. If $K/m$ has $(C, p)$-**polynomial eigenvalue decay** for $p > 1$ then $d_\eta(K_\gamma) \leq \left( \frac{C}{(p-1)\eta} \right)^{1/p} + 2.$*

*2. If $K/m$ has $C$-**exponential eigenvalue decay** then $d_\eta(K_\gamma) \leq \log \left( \frac{C}{(e-1)\eta} \right) + 2.$*

# 5 Compression Scheme

The above analysis gives us a sparse solution for the problem and, in turn, an $\epsilon$-approximation for the error on the overall sample $\mathcal{S}$ with probability $1 - \delta$. We can now fully define our compression scheme for the hypothesis class $H_\psi$ with respect to samples satisfying the eigenvalue decay property.

**Selection Scheme** $\kappa$: Given input $\mathcal{S} = (\mathbf{x}_i, y_i)_{i=1}^m$,

1. Use RLS-Nyström Sampling [28] to compute $\bar{K}_\gamma = K_\gamma S(S^T K_\gamma S)^\dagger S^T K_\gamma$ for $\eta = \frac{\epsilon^3}{5832B}$ and $\gamma = \frac{\epsilon^3}{5832Bm}$. Let $\mathcal{I}$ be the sub-sample corresponding to the columns selected using $S$.

2. Solve Optimization Problem 6 for $\lambda = \frac{\epsilon^2}{324B}$ to get $\bar{\alpha}_\gamma$.

3. Compute the $|\mathcal{I}|$-sparse vector $\alpha^* = S(S^T K_\gamma S)^\dagger S^T K_\gamma \bar{\alpha}_\gamma = K_\gamma^{-1} \bar{K}_\gamma \bar{\alpha}_\gamma$ ($K_\gamma$ is invertible as all eigenvalues are non-zero).

4. Output subsample $\mathcal{I}$ along with $\tilde{\alpha}^*$ which is $\alpha^*$ truncated to precision $\frac{\epsilon}{4M|\mathcal{I}|}$ per non-zero index.

**Reconstruction Scheme** $\rho$: Given input subsample $\mathcal{I}$ and $\tilde{\alpha}^*$, output hypothesis, $h_\mathcal{S}(\mathbf{x}) = clip_{0,1}(\mathbf{w}^T \tilde{\alpha}^*)$ where $\mathbf{w}$ is a vector with entries $K(\mathbf{x}_i, \mathbf{x}) + \gamma m \mathbb{1}[\mathbf{x} = \mathbf{x}_i]$ for $i \in \mathcal{I}$ and 0 otherwise where $\gamma = \frac{\epsilon^3}{5832Bm}$. Note, $clip_{a,b}(x) = \max(a, \min(b, x))$ for some $a < b$.

The following theorem shows that the above is a compression scheme for $\mathcal{H}_\psi$.

**Theorem 11.** $(\kappa, \rho)$ *is an $(\epsilon, \delta)$-approximate agnostic compression scheme for the hypothesis class $H_\psi$ for sample $\mathcal{S}$ of size $k(m, \epsilon, \delta, B, M) = O\left( d \log\left( \frac{d}{\delta} \right) \log\left( \frac{\sqrt{m}BMd\log(d/\delta)}{\epsilon^4} \right) \right)$ where $d$ is the $\eta$-effective dimension of $K_\gamma$ for $\eta = \frac{\epsilon^3}{5832B}$ and $\gamma = \frac{\epsilon^3}{5832Bm}$.*

# 6 Putting It All Together: From Compression to Learning

We now present our final algorithm: *Compressed Kernel Regression* (Algorithm 1). Note that the algorithm is efficient and takes at most $O(m^3)$ time.

For our learnability result, we restrict distributions to those that satisfy eigenvalue decay.

**Definition 12** (Distribution Satisfying Eigenvalue Decay). *Consider distribution $\mathcal{D}$ over $\mathcal{X}$ and kernel function $k_\psi$. Let $\mathcal{S}$ be a sample drawn i.i.d. from the distribution $\mathcal{D}$ and $K$ be the empirical gram matrix corresponding to kernel function $k_\psi$ on $\mathcal{S}$.*

*1. $\mathcal{D}$ is said to satisfy $(C, p, N)$-polynomial eigenvalue decay if with probability $1 - \delta$ over the drawn sample of size $m \geq N$, $K/m$ satisfies $(C, p)$-polynomial eigenvalue decay.*

---
**Algorithm 1** Compressed Kernel Regression
---

**Input**: Samples $\mathcal{S} = (\mathbf{x}_i, y_i)_{i=1}^m$, gram matrix $K$ on $\mathcal{S}$, constants $\epsilon, \delta > 0$, norm bound $B$ and maximum kernel function value $M$ on $\mathcal{X}$.

1: Using RLS-Nyström Sampling [28] with input $(K_\gamma, \eta m)$ for $\gamma = \frac{\epsilon^3}{5832Bm}$ and $\eta = \frac{\epsilon^3}{5832B}$ compute $\bar{K}_\gamma = K_\gamma S(S^T K_\gamma S)^\dagger S^T K_\gamma$. Let $\mathcal{I}$ be the subsample corresponding to the columns selected using $S$. Note that the number of columns selected depends on the $\eta$ effective dimension of $K_\gamma$.

2: Solve Optimization Problem 6 for $\lambda = \frac{\epsilon^2}{324B}$ to get $\bar{\alpha}_\gamma$ over $\mathcal{S}$

3: Compute $\alpha^* = S(S^T K_\gamma S)^\dagger S^T K_\gamma \bar{\alpha}_\gamma = K_\gamma^{-1} \bar{K}_\gamma \bar{\alpha}_\gamma$

4: Compute $\tilde{\alpha}^*$ by truncating each entry of $\alpha^*$ up to precision $\frac{\epsilon}{4M|\mathcal{I}|}$

**Output**: $h_{\mathcal{S}}$ such that for all $x \in \mathcal{X}$, $h_{\mathcal{S}}(\mathbf{x}) = clip_{0,1}(\mathbf{w}^T \tilde{\alpha}^*)$ where $\mathbf{w}$ is a vector with entries $K(\mathbf{x}_i, \mathbf{x}) + \gamma m \mathbb{1}[\mathbf{x} = \mathbf{x}_i]$ for $i \in \mathcal{I}$ and 0 otherwise.

---

*2. $\mathcal{D}$ is said to satisfy $(C, N)$-exponential eigenvalue decay if with probability $1 - \delta$ over the drawn sample of size $m \geq N$, $K/m$ satisfies $C$-exponential eigenvalue decay.*

Our main theorem proves generalization of the hypothesis output by Algorithm 1 for distributions satisfying eigenvalue decay in the above sense.

**Theorem 13** (Formal for Theorem 1). *Fix function class $\mathcal{C}$ with output bounded in $[0, 1]$ and $M$-bounded kernel function $k_\psi$ such that $\mathcal{C}$ is $\epsilon_0$-approximated by $H_\psi = \{\boldsymbol{x} \to \langle \boldsymbol{v}, \psi(\boldsymbol{x}) \rangle | \boldsymbol{v} \in \mathcal{K}_\psi, \langle \boldsymbol{v}, \boldsymbol{v} \rangle \leq B\}$ for some $\psi, B$. Consider a sample $\mathcal{S} = \{(\boldsymbol{x}_i, y_i)_{i=1}^m\}$ drawn i.i.d. from $\mathcal{D}$ on $\mathcal{X} \times [0, 1]$. There exists an algorithm $\mathcal{A}$ that outputs hypothesis $h_{\mathcal{S}} = \mathcal{A}(\mathcal{S})$, such that,*

*1. If $\mathcal{D}_{\mathcal{X}}$ satisfies $(C, p, m)$-polynomial eigenvalue decay with probability $1 - \delta/4$ then with probability $1 - \delta$ for $m = \tilde{O}((CB)^{1/p} \log(M) \log(1/\delta)/\epsilon^{2+3/p})$,*

$$\mathbb{E}_{(\boldsymbol{x},y) \sim \mathcal{D}}(h_{\mathcal{S}}(\boldsymbol{x}) - y)^2 \leq \min_{c \in \mathcal{C}} \left( \mathbb{E}_{(\boldsymbol{x},y) \sim \mathcal{D}}(c(\boldsymbol{x}) - y)^2 \right) + 2\epsilon_0 + \epsilon$$

*2. If $\mathcal{D}_{\mathcal{X}}$ satisfies $(C, m)$-exponential eigenvalue decay with probability $1 - \delta/4$ then with probability $1 - \delta$ for $m = \tilde{O}(\log CB \log(M) \log(1/\delta)/\epsilon^2)$,*

$$\mathbb{E}_{(\boldsymbol{x},y) \sim \mathcal{D}}(h_{\mathcal{S}}(\boldsymbol{x}) - y)^2 \leq \min_{c \in \mathcal{C}} \left( \mathbb{E}_{(\boldsymbol{x},y) \sim \mathcal{D}}(c(\boldsymbol{x}) - y)^2 \right) + 2\epsilon_0 + \epsilon$$

*Algorithm $\mathcal{A}$ runs in time* $\text{poly}(m, n)$.

**Remark:** The above theorem can be extended to different rates of eigenvalue decay. For example, for *finite* rank $r$ the obtained bound is independent of $B$ but dependent instead on $r$. Also, as in the proof of Theorem 10, it suffices for the eigenvalue decay to hold only after sufficiently large $i$.

## 7 Learning Neural Networks

Here we apply our main theorem to the problem of learning neural networks. For technical definitions of neural networks, we refer the reader to [43].

**Definition 14** (Class of Neural Networks [16]). *Let $\mathcal{N}[\sigma, D, W, T]$ be the class of fully-connected, feed-forward networks with $D$ hidden layers, activation function $\sigma$ and quantities $W$ and $T$ described as follows:*

*1. Weight vectors in layer $0$ have 2-norm bounded by $T$.*

*2. Weight vectors in layers $1, \ldots, D$ have 1-norm bounded by $W$.*

*3. For each hidden unit $\sigma(\boldsymbol{w} \cdot \boldsymbol{z})$ in the network, we have $|\boldsymbol{w} \cdot \boldsymbol{z}| \leq T$ (by $\boldsymbol{z}$ we denote the input feeding into unit $\sigma$ from the previous layer).*

We consider activation functions $\sigma_{relu}(x) = \max(0, x)$ and $\sigma_{sig} = \frac{1}{1+e^{-x}}$, though other activation functions fit within our framework. Goel et al. [16] showed that the class of ReLUs/Sigmoids along with their compositions can be approximated by linear functions in a high dimensional Hilbert space

(corresponding to a particular type of polynomial kernel). As mentioned earlier, the sample complexity of prior work depends linearly on $B$, which, for even a single ReLU, is exponential in $1/\epsilon$. Assuming sufficiently strong eigenvalue decay, we can show that we can obtain fully polynomial time algorithms for the above classes.

**Theorem 15.** *For $\epsilon, \delta > 0$, consider $\mathcal{D}$ on $\mathbb{S}^{n-1} \times [0,1]$ such that,*

*1. For $\mathcal{C}_{relu} = \mathcal{N}[\sigma_{relu}, 0, \cdot, 1]$, $\mathcal{D}_{\mathcal{X}}$ satisfies $(C, p, m)$-polynomial eigenvalue decay for $p \geq \xi/\epsilon$,*

*2. For $\mathcal{C}_{relu-D} = \mathcal{N}[\sigma_{relu}, D, W, T]$, $\mathcal{D}_{\mathcal{X}}$ satisfies $(C, p, m)$-polynomial eigenvalue decay for $p \geq (\xi W^D DT/\epsilon)^D$,*

*3. For $\mathcal{C}_{sig-D} = \mathcal{N}[\sigma_{sig}, D, W, T]$, $\mathcal{D}_{\mathcal{X}}$ satisfies $(C, p, m)$-polynomial eigenvalue decay for $p \geq (\xi T \log(W^D D/\epsilon)))^D$,*

*where $\mathcal{D}_{\mathcal{X}}$ is the marginal distribution on $\mathcal{X} = \mathbb{S}^{n-1}$, $\xi > 0$ is some sufficiently large constant and $C \leq (n \cdot 1/\epsilon \cdot \log(1/\delta))^{\zeta p}$ for some constant $\zeta > 0$. The value of $m$ is obtained from Theorem 13 as $m = \tilde{O}(C^{1/p}\epsilon^{2+3/p})$.*

*Each decay assumption above implies an algorithm for agnostically learning the corresponding class on $\mathbb{S}^{n-1} \times [0,1]$ with respect to the square loss in time $\mathsf{poly}(n, 1/\epsilon, \log(1/\delta))$.*

Note that assuming an exponential eigenvalue decay (stronger than polynomial) will result in efficient learnability for much broader classes of networks.

Since it is not known how to agnostically learn even a single ReLU with respect to arbitrary distributions on $\mathbb{S}^{n-1}$ in polynomial-time[4], much less a network of ReLUs, we state the following corollary highlighting the decay we require to obtain efficient learnability for simple networks:

**Corollary 16** (Restating Corollary 2). *Let $\mathcal{C}$ be the class of all fully-connected networks of ReLUs with one-hidden layer of size $\ell$ feeding into a final output ReLU activation where the 2-norms of all weight vectors are bounded by $1$. Then, (suppressing the parameter $m$ for simplicity), assuming $(C, i^{-\ell/\epsilon})$-polynomial eigenvalue decay for $C = \mathsf{poly}(n, 1/\epsilon, \ell)$, $\mathcal{C}$ is learnable in polynomial time with respect to square loss on $\mathbb{S}^{n-1}$. If ReLU is replaced with sigmoid, then we require eigenvalue decay of $i^{-\sqrt{\ell}\log(\sqrt{\ell}/\epsilon)}$.*

# 8 Conclusions and Future Work

We have proposed the first set of distributional assumptions that guarantee fully polynomial-time algorithms for learning expressive classes of neural networks (without restricting the structure of the network). The key abstraction was that of a *compression scheme* for kernel approximations, specifically Nyström sampling. We proved that eigenvalue decay of the Gram matrix reduces the dependence on the norm $B$ in the kernel regression problem.

Prior distributional assumptions, such as the underlying marginal equaling a Gaussian, neither lead to fully polynomial-time algorithms nor are representative of real-world data sets[5]. Eigenvalue decay, on the other hand, has been observed in practice and does lead to provably efficient algorithms for learning neural networks.

A natural criticism of our assumption is that the rate of eigenvalue decay we require is too strong. In some cases, especially for large depth networks with many hidden units, this may be true[6]. Note, however, that our results show that even moderate eigenvalue decay will lead to improved algorithms. Further, it is quite possible our assumptions can be relaxed. An obvious question for future work is what is the minimal rate of eigenvalue decay needed for efficient learnability? Another direction would be to understand how these eigenvalue decay assumptions relate to other distributional assumptions.

**Acknowledgements.** We would like to thank Misha Belkin and Nikhil Srivastava for very helpful conversations regarding kernel ridge regression and eigenvalue decay. We also thank Daniel Hsu, Karthik Sridharan, and Justin Thaler for useful feedback. The analogy between eigenvalue decay and power-law graphs is due to Raghu Meka.

## Footnotes

[4]Goel et al. [16] show that agnostically learning a single ReLU over $\{-1, 1\}^n$ is as hard as learning sparse parities with noise. This reduction can be extended to the case of distributions over $\mathbb{S}^{n-1}$ [3].

[5]Despite these limitations, we still think uniform or Gaussian assumptions are worthwhile and have provided highly nontrivial learning results.

[6]It is useful to keep in mind that agnostically learning even a single ReLU with respect to all distributions seems computationally intractable, and that our required eigenvalue decay in this case is only a function of the accuracy parameter $\epsilon$.

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
