[Supplementary Material]

# Supplemental: Eigenvalue Decay Implies Polynomial-Time Learnability for Neural Networks

**Surbhi Goel** [*]
Department of Computer Science
University of Texas at Austin
surbhi@cs.utexas.edu

**Adam Klivans** [†]
Department of Computer Science
University of Texas at Austin
klivans@cs.utexas.edu

## A  Background on Learning Models and Generalization Bounds

### A.1  Model and Generalization Bounds

We will work in the general non-realizable model of statistical learning theory also known as the *agnostic model of learning*. In this model, the labels presented to the learner are arbitrary, and the goal is to output a hypothesis that is competitive with the best fitting function from some fixed class:

**Definition A** (Agnostic Learning [5, 3]). *A concept class $\mathcal{C} \subseteq \mathcal{Y}^{\mathcal{X}}$ is agnostically learnable with respect to loss function $l : \mathcal{Y}' \times \mathcal{Y} \to \mathbb{R}^+$ (where $\mathcal{Y} \subseteq \mathcal{Y}'$) and distribution $D$ over $\mathcal{X} \times \mathcal{Y}$, if for every $\delta, \epsilon > 0$ there exists a learning algorithm $\mathcal{A}$ given access to examples drawn from $D$, $\mathcal{A}$ outputs a hypothesis $h : \mathcal{X} \to \mathcal{Y}'$, such that with probability at least $1 - \delta$,*

$$E_{(\boldsymbol{x},y) \sim D}[l(h(\boldsymbol{x}), y)] \leq \min_{c \in C} E_{(\boldsymbol{x},y) \sim D}[l(c(\boldsymbol{x}), y)] + \epsilon. \tag{1}$$

*Furthermore, we say that $\mathcal{C}$ is* efficiently agnostically learnable to error $\epsilon$ *if $\mathcal{A}$ can output an $h$ satisfying Equation* (1) *with running time polynomial in $n$, $1/\epsilon$ and $1/\delta$.*

The agnostic model generalizes Valiant's PAC model of learning [6], and so all of our results will hold for PAC learning as well. The following is a well known theorem for proving generalization based on Rademacher complexity.

**Theorem A** ([1]). *Let $\mathcal{D}$ be a distribution over $\mathcal{X} \times \mathcal{Y}$ and let $l : \mathcal{Y}' \times \mathcal{Y}$ be a b-bounded loss function that is L-Lispschitz in its first argument. Let $\mathcal{F}$ be a class of functions from $\mathcal{X}$ to $\mathcal{Y}'$ and for any $f \in \mathcal{F}$, and $S = ((\boldsymbol{x}_1, y_1), \ldots, (\boldsymbol{x}_m, y_m)) \sim \mathcal{D}^m$ and $\delta > 0$, with probability at least $1 - \delta$ we have,*

$$\left| E_{(x,y) \sim \mathcal{D}}[l(f(\boldsymbol{x}), y)] - \frac{1}{m} \sum_{i=1}^{m} l(f(\boldsymbol{x}_i), y_i) \right| \leq 4 \cdot L \cdot \mathcal{R}_m(\mathcal{F}) + 2 \cdot b \cdot \sqrt{\frac{\log(1/\delta)}{2m}}$$

*where $\mathcal{R}_m(\mathcal{F})$ is the Rademacher complexity of the function class $\mathcal{F}$.*

The Rademacher complexity of this linear class can be bounded by using the following theorem.

**Theorem B** ([4]). *Let $\mathcal{K}$ be a subset of a Hilbert space equipped with inner product $\langle \cdot, \cdot \rangle$ such that for each $x \in \mathcal{K}$, $\langle \boldsymbol{x}, \boldsymbol{x} \rangle \leq X^2$, and let $\mathcal{W} = \{\boldsymbol{x} \to \langle \boldsymbol{x}, \boldsymbol{w} \rangle \mid \langle \boldsymbol{w}, \boldsymbol{w} \rangle \leq W^2\}$ be a class of linear functions. Then it holds that*

$$\mathcal{R}_m(\mathcal{W}) \leq X \cdot W \cdot \sqrt{\frac{1}{m}}.$$

---

[*]Work supported by a Microsoft Data Science Initiative Award.

[†]Part of this work was done while visiting the Simons Institute for Theoretical Computer Science.

# B  Proof of Theorem 8

We bound the error for each of the approximations: sparsification, preconditioning and lagrangian relaxation in the following lemma.

**Lemma A.** *The errors due to the following approximations can be bounded as follows.*

1. *Error due to sparsification:* $||\bar{K}_\gamma \bar{\alpha}_\gamma - Y||_2 \leq ||K_\gamma \alpha_\gamma - Y||_2 + \frac{\eta\sqrt{m}}{\lambda+\gamma}$

2. *Error due to preconditioning:* $||K_\gamma \alpha_\gamma - Y||_2 \leq ||K\alpha - Y||_2 + \frac{\gamma\sqrt{m}}{\lambda+\gamma}$

3. *Error due to lagrangian relaxation:* $||K\alpha - Y||_2 \leq ||K\alpha_B - Y||_2 + \sqrt{\lambda m B}$

*Proof.* The errors can be bounded as follows.

1. We have,

$$||\bar{K}_\gamma \bar{\alpha}_\gamma - Y||_2 - ||K_\gamma \alpha_\gamma - Y||_2$$

$$\leq ||\bar{K}_\gamma \bar{\alpha}_\gamma - K_\gamma \alpha_\gamma||_2 \tag{2}$$

$$= ||\bar{K}_\gamma \left(\bar{K}_\gamma + \lambda m I\right)^{-1} Y - K_\gamma \left(K_\gamma + \lambda m I\right)^{-1} Y||_2 \tag{3}$$

$$= \lambda m || \left(- \left(\bar{K}_\gamma + \lambda m I\right)^{-1} + \left(K_\gamma + \lambda m I\right)^{-1}\right) Y||_2 \tag{4}$$

$$= \lambda m || \left(\bar{K}_\gamma + \lambda m I\right)^{-1} \left(\bar{K}_\gamma - K_\gamma\right) \left(K_\gamma + \lambda m I\right)^{-1} Y||_2 \tag{5}$$

$$\leq \lambda m || \left(\bar{K}_\gamma + \lambda m I\right)^{-1} ||_2 ||\bar{K}_\gamma - K_\gamma||_2 || \left(K + (\lambda+\gamma)m I\right)^{-1} ||_2 ||Y||_2 \tag{6}$$

$$\leq \frac{||\bar{K}_\gamma - K_\gamma||_2}{(\lambda+\gamma)\sqrt{m}} \leq \frac{\eta\sqrt{m}}{\lambda+\gamma}. \tag{7}$$

Here 2 follows from triangle inequality, 3 follows from substitution and 4 follows from using $A\left(A+cI\right)^{-1} = \left(A+cI-cI\right)\left(A+cI\right)^{-1} = I - c\left(A+cI\right)^{-1}$. 5 follows from $a^{-1}-b^{-1} = -a^{-1}\left(a-b\right)b^{-1}$ and 6 follows from $||AB||_2 \leq ||A||_2||B||_2$. Lastly 7 follows from $||A^{-1}||_2 = \lambda_{min}\left(A\right)^{-1}$, $\lambda_{min}\left(A+cI\right) \geq c$ for psd $A$. We also use $K_\gamma = K + \gamma m I$ and $||Y||_2 \leq \sqrt{m}$.

2. Similar to the above proof, we have,

$$||K_\gamma \alpha_\gamma - Y||_2 - ||K\alpha - Y||_2$$

$$\leq ||K_\gamma \alpha_\gamma - K(K + \lambda m I)^{-1} Y||_2 \tag{8}$$

$$= ||K_\gamma \left(K_\gamma + \lambda m I\right)^{-1} Y - K \left(K + \lambda m I\right)^{-1} Y||_2 \tag{9}$$

$$= \lambda m || \left(K_\gamma + \lambda m I\right)^{-1} \left(K_\gamma - K\right) \left(K + \lambda m I\right)^{-1} Y||_2 \tag{10}$$

$$\leq \lambda m || \left(K + (\lambda+\gamma)m I\right)^{-1} ||_2 ||\gamma m I||_2 || \left(K + \lambda m I\right)^{-1} ||_2 ||Y||_2 \tag{11}$$

$$\leq \frac{\gamma\sqrt{m}}{\lambda+\gamma}. \tag{12}$$

3. Since $\alpha$ minimizes Optimization Problem 4, we have

$$||K\alpha - Y||_2^2 \leq ||K\alpha - Y||_2^2 + \lambda m \alpha^T K \alpha \tag{13}$$

$$\leq ||K\alpha_B - Y||_2^2 + \lambda m \alpha_B^T K \alpha_B \tag{14}$$

$$\leq ||K\alpha_B - Y||_2^2 + \lambda m B \tag{15}$$

where the last inequality follows from $\alpha_B^T K \alpha_B \leq B$ by the constraint of the bounded optimization problem. Taking the square-root, we get,

$$||K\alpha - Y||_2 \leq \sqrt{||K\alpha_B - Y||_2^2 + \lambda m B} \leq ||K\alpha_B - Y||_2 + \sqrt{\lambda m B} \tag{16}$$

$\square$

Note that $\bar{K}\bar{\alpha}_\gamma = K_\gamma \alpha^*$ by the definition of $\alpha^*$, from the previous lemma, we have,

$$||\bar{K}\bar{\alpha}_\gamma - Y||_2 - ||K\alpha_B - Y||_2 \leq \frac{\eta\sqrt{m}}{\lambda + \gamma} + \frac{\gamma\sqrt{m}}{\lambda + \gamma} + \sqrt{\lambda m}B = \beta \tag{17}$$

where $\beta = \frac{(\eta+\gamma)\sqrt{m}}{\lambda+\gamma} + \sqrt{\lambda m}B$. Squaring and then dividing by $m$ on both sides, we get

$$\frac{1}{m}||\bar{K}_\gamma\bar{\alpha}_\gamma - Y||_2^2 \leq \frac{1}{m}||K\alpha_B - Y||_2^2 + 2\frac{\beta}{m}||K\alpha_B - Y||_2 + \frac{\beta^2}{m} \tag{18}$$

$$\leq \frac{1}{m}||K\alpha_B - Y||_2^2 + 2\frac{\beta}{\sqrt{m}} + \frac{\beta^2}{m} \tag{19}$$

$$\leq \frac{1}{m}||K\alpha_B - Y||_2^2 + 3\frac{\beta}{\sqrt{m}} \tag{20}$$

The second inequality follows from $||K\alpha_B - Y||_2^2 \leq ||Y||_2^2 \leq m$ since $0$ is a feasible solution for Optimization Problem 3. The last inequality follows from assuming $\frac{\beta}{\sqrt{m}} \leq 1$ which holds for our choice of $\beta$. Setting the values in the lemma satisfies the last inequality gives us $\beta \leq \frac{\epsilon\sqrt{m}}{3}$ giving us the desired bound.

## C    Proof of Theorem 10

Observe that,

$$d_\eta(K_\gamma) = tr(K_\gamma(K_\gamma + \eta m I)^{-1})$$

$$= \sum_{i=1}^{m} \frac{\lambda_i(K_\gamma)}{\lambda_i(K_\gamma) + \eta m}$$

$$\leq \sum_{i=1}^{j} \frac{\lambda_i(K_\gamma)}{\lambda_i(K_\gamma)} + \sum_{i=j+1}^{m} \frac{\lambda_i(K_\gamma)}{\eta m}$$

$$\leq j + \sum_{i=j+1}^{m} \frac{\gamma m + \lambda_i(K)}{\eta m}$$

$$\leq j + 1 + \sum_{i=j+1}^{m} \frac{\lambda_i(K)}{\eta m}$$

Here the second equality follows from trace of matrix being equal to the sum of the eigenvalues and the last follows from $\gamma m \leq \eta$.

1.  For $(C, p)$-polynomial eigenvalue decay with $p > 1$,

$$\sum_{i=k+1}^{m} \frac{\lambda_i(K)}{\eta m} = \sum_{i=k+1}^{m} \frac{Ci^{-p}}{\eta} \leq \frac{C}{\eta} \int_{k+1}^{\infty} i^{-p} di = \frac{C(k+1)^{-p+1}}{(p-1)\eta}$$

    Substituting $j = \left(\frac{C}{(p-1)\eta}\right)^{1/p}$ we get the required bound.

2.  For $C$-exponential eigenvalue decay,

$$\sum_{i=k+1}^{m} \frac{\lambda_i(K)}{\eta m} = \sum_{i=k+1}^{m} \frac{Ce^{-i}}{\eta} \leq \sum_{i=k+1}^{\infty} \frac{Ce^{-i}}{\eta} = \frac{Ce^{-k}}{(e-1)\eta}$$

    Substituting $j = \log\left(\frac{C}{(e-1)\eta}\right)$ we get the required bound.

**Remark**: *Based on the above analysis, observe that we only need the eigenvalue decay to hold after the $j$th eigenvalue for $j$ defined above. Thus the top $j-1$ eigenvalues need not be constrained.*

# D Proof of Theorem 11

For $\mathcal{S} = (\mathbf{x}_i, y_i)_{i=1}^m$ and $h_\mathcal{S}$ the output of the compression scheme, we have

$$\frac{1}{m}\sum_{i=1}^m (h_\mathcal{S}(\mathbf{x}_i) - y_i)^2 \leq \frac{1}{m}\sum_{i=1}^m \left(\sum_{j\in\mathcal{I}}(K(\mathbf{x}_j, \mathbf{x}_i) + \gamma m\mathbb{1}[\mathbf{x}_j = \mathbf{x}_i])\tilde{\alpha}_j^* - y_i\right)^2 \tag{21}$$

$$\leq \frac{1}{m}\sum_{i=1}^m \left(\sum_{j\in\mathcal{I}}(K(\mathbf{x}_j, \mathbf{x}_i) + \gamma m\mathbb{1}[\mathbf{x}_j = \mathbf{x}_i])\alpha_j^* - y_i\right)^2 + \frac{\epsilon}{2} \tag{22}$$

$$= \frac{1}{m}\|K_\gamma\alpha^* - Y\|_2^2 + \frac{\epsilon}{2} \tag{23}$$

$$= \frac{1}{m}\|\bar{K}_\gamma\bar{\alpha}_\gamma - Y\|_2^2 + \frac{\epsilon}{2} \tag{24}$$

$$= \frac{1}{m}\|K\alpha_B - Y\|_2^2 + \frac{\epsilon}{2} + \frac{\epsilon}{2} \tag{25}$$

$$= \min_{h\in H_\psi}\left(\frac{1}{m}\sum_{i=1}^m (h(\mathbf{x}_i) - y_i)^2\right) + \epsilon \tag{26}$$

Here 21 follows from the fact that since the output is in $[0,1]$ clipping only reduces the loss, 22 follows from the precision used while compressing and since square loss is 2-Lipschitz, 23 follows from representing it in the matrix form, 24 follows since $\alpha^* = K_\gamma^{-1}\bar{K}_\gamma\bar{\alpha}_\gamma$ by definition, 25 follows from Theorem 8 with the given parameters satisfying the theorem for $\epsilon/2$ and lastly 26 follows from the definition of $\alpha_B$.

The size of the above scheme can be bounded using the following lemma.

**Lemma B.** *The bit complexity of the side information of the selection scheme $\kappa$ given above is* $O\left(d\log\left(\frac{d}{\delta}\right)\log\left(\frac{\sqrt{m}BMd\log(d/\delta)}{\epsilon^4}\right)\right)$ *where $d$ is the $\eta$-effective dimension of $K_\gamma$ for $\eta = \frac{\epsilon^3}{5832B}$ and* $\gamma = \frac{\epsilon^3}{5832Bm}$.

*Proof.* From the selection scheme we can bound the norm of $\alpha^* = K_\gamma^{-1}\bar{K}_\gamma\bar{\alpha}_\gamma$ for $\gamma = \frac{\epsilon^3}{5832Bm}$, the side information, as follows,

$$\|\alpha^*\|_2 = \|K_\gamma^{-1}\bar{K}_\gamma\bar{\alpha}_\gamma\|_2 \tag{27}$$

$$= \|K_\gamma^{-1}\bar{K}_\gamma(\bar{K}_\gamma + \lambda mI)^{-1}Y\|_2 \tag{28}$$

$$\leq \|K_\gamma^{-1}\|_2\|\bar{K}_\gamma(\bar{K}_\gamma + \lambda mI)^{-1}\|_2\|Y\|_2 \tag{29}$$

$$\leq \frac{1}{\gamma m}\cdot 1 \cdot \sqrt{m} \tag{30}$$

$$= \frac{1}{\gamma\sqrt{m}} = \frac{5832\sqrt{m}B}{\epsilon^3}. \tag{31}$$

Thus we can upper bound the bit complexity of the non-decimal part of $\alpha^*$ as,

$$\sum_{i\in\mathcal{I}}\log(|\alpha_i^*|) = \frac{1}{2}\sum_{i=1}^{|\mathcal{I}|}\log\left((\alpha_i^*)^2\right)$$

$$\leq \frac{|\mathcal{I}|}{2}\log\left(\frac{\sum_{i=1}^{|\mathcal{I}|}(\alpha_i^*)^2}{|\mathcal{I}|}\right)$$

$$\leq |\mathcal{I}|\log\left(\frac{\|\alpha^*\|_2}{\sqrt{|\mathcal{I}|}}\right) \leq |\mathcal{I}|\log\left(\frac{5832\sqrt{m}B}{\epsilon^3}\right)$$

where $|\mathcal{I}| = O\left(d\log\left(\frac{d}{\delta}\right)\right)$ according to Theorem 7. Since each non-zero index has $\frac{\epsilon}{4M|\mathcal{I}|}$ precision, we need $|\mathcal{I}|\log\left(\frac{4M|\mathcal{I}|}{\epsilon}\right)$ bits for the decimal part. Combining the two-parts we get the required bound. $\qquad\square$

# E  Proof of Theorem 13

Since $\mathcal{C}$ is $\epsilon_0$-approximated by $H_\psi$ we have,

$$\min_{h \in H_\psi} \left( \frac{1}{m} \sum_{i=1}^{m} (h(\mathbf{x}_i) - y_i)^2 \right) \leq \min_{c \in \mathcal{C}} \left( \frac{1}{m} \sum_{i=1}^{m} (c(\mathbf{x}_i) - y_i)^2 \right) + 2\epsilon_0 \leq \frac{1}{m} \sum_{i=1}^{m} (c^*(\mathbf{x}_i) - y_i)^2 + 2\epsilon_0$$

where $c^* \in \mathcal{C}$ be such that it minimizes $\mathbb{E}_{(x,y) \sim \mathcal{D}} (c(x) - y)^2$ over all $c \in \mathcal{C}$. The first inequality follows from square loss being 2-Lipschitz and the last inequality follows from $c^*$ being a feasible solution.

Let $K$ be the empirical gram matrix corresponding to $k_\psi$ on $\mathcal{S}$. Let $h_\mathcal{S}$ be the hypothesis output by Algorithm 1 with input $(\mathcal{S}, K, \epsilon_1, \delta/4, B, M)$ for $\epsilon_1 > 0$ chosen later. From Theorem 11 with probability $1 - \delta/4$, we have

$$\frac{1}{m} \sum_{i=1}^{m} (h_\mathcal{S}(\mathbf{x}_i) - y_i)^2 \leq \min_{h \in H_\psi} \left( \frac{1}{m} \sum_{i=1}^{m} (h(\mathbf{x}_i) - y_i)^2 \right) + \epsilon_1.$$

We know that for every $c \in \mathcal{C}$, the square loss is bounded by 1, thus using Chernoff-Hoeffding inequality, with probability $1 - \delta/4$, we have

$$\frac{1}{m} \sum_{i=1}^{m} (c^*(\mathbf{x}_i) - y_i)^2 \leq \mathbb{E}_{(\mathbf{x},y) \sim \mathcal{D}} (c^*(\mathbf{x}) - y)^2 + \epsilon_2$$

where $\epsilon_2 = \sqrt{\frac{\log(4/\delta)}{2m}}$.

Now the output of $h_\mathcal{S}$ lies in $[0, 1]$ thus for all $(\mathbf{x}, y)$, $(y - h_\mathcal{S}(\mathbf{x}))^2$ lies in $[0, 1]$. Thus viewing $h_\mathcal{S}$ as the output of the compression scheme $(\kappa, \rho)$ of size $k$ (Theorem 11), by Theorem 4, we have with probability $1 - \delta/4$,

$$\left| \mathbb{E}_{(\mathbf{x},y) \sim \mathcal{D}} (h_\mathcal{S}(\mathbf{x}) - y)^2 - \frac{1}{m} \sum_{i=1}^{m} (h_S(\mathbf{x}_i) - y_i)^2 \right| \leq \sqrt{\frac{\epsilon_3}{m} \sum_{i=1}^{m} (h_\mathcal{S}(\mathbf{x}_i) - y_i)^2} + \epsilon_3 \leq \epsilon_3 + \sqrt{\epsilon_3} \leq 2\sqrt{\epsilon_3}$$

where $\epsilon_3 = 50 \cdot \frac{k \log(m/k) + \log(4/\delta)}{m}$.

Combining the above, we have with probability $1 - \delta$,

$$\mathbb{E}_{(\mathbf{x},y) \sim \mathcal{D}} (h_\mathcal{S}(\mathbf{x}) - y)^2 \leq \frac{1}{m} \sum_{i=1}^{m} (h_S(\mathbf{x}_i) - y_i)^2 + 2\sqrt{\epsilon_3} \tag{32}$$

$$\leq \min_{h \in H_\psi} \left( \frac{1}{m} \sum_{i=1}^{m} (h(\mathbf{x}_i) - y_i)^2 \right) + \epsilon_1 + 2\sqrt{\epsilon_3} \tag{33}$$

$$\leq \frac{1}{m} \sum_{i=1}^{m} (c^*(\mathbf{x}_i) - y_i)^2 + 2\epsilon_0 + \epsilon_1 + 2\sqrt{\epsilon_3} \tag{34}$$

$$\leq \min_{c \in \mathcal{C}} \left( \mathbb{E}_{(\mathbf{x},y) \sim \mathcal{D}} (c(\mathbf{x}) - y)^2 \right) + 2\epsilon_0 + \epsilon_1 + \epsilon_2 + 2\sqrt{\epsilon_3} \tag{35}$$

Using Theorem 10 we can bound $k$ depending on the different eigenvalue decay assumption. Now we set $\epsilon_1 = \epsilon/3$ and substituting for $m$. Recall that $\epsilon_2$ and $\epsilon_3$ are functions of $m$ and for the chosen $m$, they are bounded by $\epsilon/3$ giving us the desired bound. Since Algorithm 1 runs in time $\text{poly}(\mathsf{m}, \mathsf{n})$ we get the required time complexity.

# F  Proof of Theorem 15

We use the following theorem that follows directly from the structural results in [2] (and uses the composed-kernel technique of Zhang et al. [7]).

**Theorem C.** *Consider the following hypothesis class* $\mathcal{H}_{\mathsf{MK}_d} = \{x \to \langle v, \psi(x) \rangle | v \in \mathcal{K}_{\mathsf{MK}_d}, \langle v, v \rangle \le B\}$ *where* $\mathcal{K}_{\mathsf{MK}_d}$ *is the Hilbert space corresponding to the Multinomial Kernel* [3] *and* $\psi$ *is the corresponding feature vector. For* $D > 0$, *consider the composed class* $\mathcal{H}^{(D)} = \{x \to \langle v, \psi^{(D)}(x) \rangle | v \in \mathcal{K}^{(D)}, \langle v, v \rangle \le B\}$ *where* $\psi^{(D)}$ *is the feature vector of the $D$-times composed kernel* $K^{(D)}$ [4]. *Then for* $\mathcal{X} = \mathbb{S}^{n-1}$,

1. ***Single ReLU***: $\mathcal{C}_{relu} = \mathcal{N}[\sigma_{relu}, 0, \cdot, 1]$ *is $\epsilon$-approximated by* $\mathcal{H}_d$ *for* $d = O(1/\epsilon)$ *and* $B = 2^{(\tau/\epsilon)}$ *with* $M = d + 1$,

2. ***Network of ReLUs***: $\mathcal{C}_{relu-D} = \mathcal{N}[\sigma_{relu}, D, W, T]$ *is $\epsilon$-approximated by* $\mathcal{H}_{(D)}$ *for* $B = 2^{(\tau W^D DT/\epsilon)^D}$ *with* $M = 2$,

3. ***Network of Sigmoids***: $\mathcal{C}_{sig-D} = \mathcal{N}[\sigma_{sig}, D, W, T]$ *is $\epsilon$-approximated by* $\mathcal{H}_{(D)}$ *for* $B = 2^{(\tau T \log(W^D D/\epsilon))^D}$ *with* $M = 2$,

*for some sufficiently large constant* $\tau > 0$.

The proof follows from applying Theorem 13 to the appropriate kernel from previous theorem and substituting the corresponding eigenvalue decays to compute the sample size needed by Algorithm 1 for learnability. For example, for the case of single ReLU, $M = \mathsf{poly}(1/\epsilon)$, $B = 2^{(\tau/\epsilon)}$ and we take $p \ge \xi/\epsilon$. So for any $C = (n \cdot 1/\epsilon)^{\zeta p}$, we obtain sample complexity $m = \tilde{O}((C2^{(\tau/\epsilon)})^{1/p} \log(M)/\epsilon^{2+3/p}) = \mathsf{poly}(n, 1/\epsilon)$. Since the algorithm takes time at most $\mathsf{poly}(m, n)$, we obtain the required result.

## G   Proof of Corollary 16

By assumption the 2-norm of each weight vector is bounded by 1, which implies that the 1-norm of the weight vector to the one hidden unit at layer two is at most $\sqrt{\ell}$. Also observe that, the maximum 2-norm of any input vector $\mathbf{z}$ to a hidden unit with weight vector $\mathbf{w}$ is bounded by $\sqrt{\ell}$ hence $|\mathbf{w} \cdot \mathbf{x}| \le \sqrt{\ell}$. Using these properties we can apply Theorem 15 with parameters $W = \sqrt{\ell}$, $T = \sqrt{\ell}$ and $D = 1$ to obtain the required result.

## Footnotes

[3]The multinomial kernel defined by [2] is $\mathsf{MK}_d(\mathbf{x}, \mathbf{x}') = \sum_{i=0}^{d} (\mathbf{x} \cdot \mathbf{x}')^i$.

[4][7] defined kernel $K^{(1)}(\mathbf{x}, \mathbf{x}') = \frac{1}{2-(\mathbf{x} \cdot \mathbf{x}')}$. The corresponding composed kernel function is defined as $K^{(D)}(\mathbf{x}, \mathbf{x}') = \frac{1}{2-K^{(D-1)}(\mathbf{x}, \mathbf{x}')}$