[Reviews · NeurIPS 2017]

Reviewer 1



Overview: The paper deals with a class of generalization bounds for kernel-based (potentially improper learning), when the kernel matrix is "approximately low rank". More precisely, if the the kernel matrix satisfies a decay of the eigenvalues, one can "compress" it by subsampling a small number of the columns, and solving an optimization problem over the smaller subset of columns. This allows the a more efficient concentration bound for proving generalization. Finally, the methods are applied to depth-2 neural networks. Comments on decision: while the result is interesting, and I applaud the direction of making assumptions on the data-distribution, which seems crucial for proving interesting result for scenarios in which neural networks work -- I find the current results in the paper too preliminary and essentially, direct in light of prior work. In particular, the relationship between compression and generalization is known (e.g. David et al. '16), as are the subsampling results when eigenvalue decay holds. This makes the general kernel learning result a "stitching" of prior work. The neural network results again follow from approximation results in Goel et al. -- so are not terribly new. (Additionally, it looks like they were written last-minute -- the bounds are only worked out for depth-2 networks. )

Reviewer 2



This paper studies the problem of learning concepts that can be approximated in a Reproducing Kernel Hilbert Space (RKHS) with bounded norm (say norm at most B, which will appear in the complexity bounds). Previous results have shown sample complexity bounds that are polynomial in B for learning neural networks, and Shamir had shown a lower bound of Omega(B) for various problems. A main contribution of this paper is to is to show that when the eigen values of the Gram matrix decay sufficiently fast, the sample complexity can be sub-linear, and even logarithmic in B. As an application they show that for data distributions that satisfy the polynomial decay, better sample complexity results can be obtained for relu's and sigmoid networks of depth one. The argument is that when the Gram matrix has fast eigen-decay, it has a small effective dimension, allowing for better compression, which in their case simply means that it can be approximated using a few columns. They use a series of reductions, each explained clearly, to obtain the compression scheme, and the complexity guarantees. - L169: I think the subscripts are mixed-up. - Can one say something by directly compressing the Gram-matrix, instead of sampling (say using PCA).

Reviewer 3



This paper considers the problem of square-loss regression for several function classes computed by neural networks. In contrast to what previous hardness results might suggest, they show that those function classes can be improperly learnt under marginal distribution assumptions alone. Specifically, they use previous structural results of approximate RKHS embedding and assume an eigenvalue decay assumption on the empirical gram matrix (namely, it is approximately low-rank). They then apply a recursive Nystrom sampling method to obtain an approximate and compressed version of the gram matrix, which is then used instead in the learning. The error of the resulting regression is then bounded using known compression generalization bounds, hence obtaining generalization bounds for kernelized regression using Compression Schemes. These in turn are claimed to be tighter than previously used bounds in terms of the required norm regularization (denoted B) in the RKHS. Under the above assumption, this leads to more efficient learning. Despite the numerous typos and unclear sentences (see below for specific comments), the paper can be well understood for the most part and I found it quite interesting. It non-trivially combines several known results to obtain very nice conclusions about a still not well understood topic. The main technical contribution is obtaining generalization bounds for kernelized regression using Compression Schemes. As the authors comment, it is indeed interesting how compression schemes can be used to improve generalization bounds for other function classes. One such work is [KSW], showing that compression-based generalization bounds similar to those considered here can be used to obtain a computationally efficient algorithm for computing a Bayes consistent 1-NN classifier in general metric space. To be more certain about my review please reply to the following: - It is claimed that the proposed algorithm has fully-polynomial time complexity in all the relevant parameters. Unfortunately, this dependence is not transparent and is implicit in the results, in particular on its dependence on B. It is only claimed in Thm. 1 that the runtime is O(log B). Please also discuss why a linear dependence in B is considered not fully polynomial. - The dependence on B is Section 7 is hidden, making it very hard to follow as how the new compression bound comes into play. The proof in the supplementary doesn't help much either. As this is the main advantage, it should be much more transparent. Specific comments: - Line 38: "fully polynomial-time algorithms are intractable" is self contradicting - Line 42: "... are necessary for efficient learnability" - I believe you should add "by gradient based algorithms". Otherwise, the next sentence in the paper is a contradiction. - A better connection to the results in [Zhang] is needed. - Thm 1.: the role of \eps should be introduced, even if informally. - Line 62: Paragraph is unclear at first reading. Better to rephrase. - Line 78: missing "with high probability" at the end of sentence? - Paragraph starting at line 79 is highly unclear. E.g. "compress the sparsity" makes no sense. "bit complexity" is unclear. - Definition 1: Unclear to what "with probability 1-\delta" refers to. Is it to property P or the empirical error bound. Missing comma? What is the property P you actually use in your proofs? - Thm. 2 has many typos. In addition, missing iid assumption. Is there no assumption on the loss function l? More importantly, in the refereed papers, only one direction of the bound is proved, as opposed to the bidirectional (i.e. having absolute value on the lhs of the bound) you give here. - Definition 2 has many typos. - Line 74: "Let us say" -> Suppose?; Also eps -> (eps,m)? - Line 187: "improve the dependence on sqrt(m)". Please add reference. - The sentence beginning in line 292 is highly unclear. - Many typos all over. - The bibliography misses many entries (year, journal, etc.). References: [KSW] - Kontorovich, Sabato and Weiss. Nearest-Neighbor Sample Compression: Efficiency, Consistency, Infinite Dimensions. 2017. [Zhang] - Tong Zhang. Effective dimension and generalization of kernel learning. 2013.